# Tinnitus Perception in Light of a Parietal Operculo–Insular Involvement: A Review

**DOI:** 10.3390/brainsci12030334

**Published:** 2022-03-01

**Authors:** Chloé Jaroszynski, Agnès Job, Maciej Jedynak, Olivier David, Chantal Delon-Martin

**Affiliations:** 1University Grenoble Alpes, Inserm, U1216, Grenoble Institut Neurosciences, 38000 Grenoble, France; chloe.jaroszynski@univ-grenoble-alpes.fr (C.J.); maciej.jedynak@univ-grenoble-alpes.fr (M.J.); olivier.david@inserm.fr (O.D.); 2Institut de Recherche Biomédicale des Armées, IRBA, 91220 Brétigny-sur-Orge, France; ajob.crssa@gmail.com; 3Aix Marseille University, Inserm, INS, Inst Neurosci Syst, 13005 Marseille, France

**Keywords:** tinnitus, parietal operculum, OP3, insula, integration area, neuroimaging, F-Tract

## Abstract

In tinnitus literature, researchers have increasingly been advocating for a clearer distinction between tinnitus perception and tinnitus-related distress. In non-bothersome tinnitus, the perception itself can be more specifically investigated: this has provided a body of evidence, based on resting-state and activation fMRI protocols, highlighting the involvement of regions outside the conventional auditory areas, such as the right parietal operculum. Here, we aim to conduct a review of available investigations of the human parietal operculo–insular subregions conducted at the microscopic, mesoscopic, and macroscopic scales arguing in favor of an auditory–somatosensory cross-talk. Both the previous literature and new results on functional connectivity derived from cortico–cortical evoked potentials show that these subregions present a dense tissue of interconnections and a strong connectivity with auditory and somatosensory areas in the healthy brain. Disrupted integration processes between these modalities may thus result in erroneous perceptions, such as tinnitus. More precisely, we highlight the role of a subregion of the right parietal operculum, known as OP3 according to the Jülich atlas, in the integration of auditory and somatosensory representation of the orofacial muscles in the healthy population. We further discuss how a dysfunction of these muscles could induce hyperactivity in the OP3. The evidence of direct electrical stimulation of this area eliciting auditory hallucinations further suggests its involvement in tinnitus perception. Finally, a small number of neuroimaging studies of therapeutic interventions for tinnitus provide additional evidence of right parietal operculum involvement.

## 1. Introduction

Tinnitus, the chronic perception of a phantom sound, is a public health issue estimated to have up to a 15% prevalence in the adult population [1,2] with 1% to 2% of the population suffering from unremitting tinnitus [3].

A recent international multidisciplinary group proposes to distinguish between *tinnitus* and *tinnitus disorders* where *tinnitus* describes “the conscious awareness of a tonal or composite noise for which there is no identifiable corresponding external acoustic source”, corresponding to the percept per se, leading to *tinnitus disorders* “when associated with emotional distress, cognitive dysfunction, and/or autonomic arousal, leading to behavioral changes and functional disability” [4]. Indeed, most tinnitus studies are prone to confounding biases related to the difficulty in disambiguating effects attributable to the presence of tinnitus perception per se from those related to comorbidities such as hearing loss, a reduced sound tolerance threshold, anxiety disorders, or an impaired quality of life often experienced by tinnitus sufferers. Facing heterogeneity of tinnitus, authors now recommend focusing studies on tinnitus sub-types, in terms of etiology (for instance resulting from noise exposure or from head or neck injuries), in terms of chronicity (less than 3 months duration for acute tinnitus or above for chronic tinnitus) [5,6], or to focus on mild forms of tinnitus whereby comorbidities may be strongly limited. In this review, we attempted to focus on tinnitus without associated disorders despite the heterogeneity of the recruitment reported in the literature.

Since tinnitus is perceived as an auditory phenomenon, the auditory pathway was historically the main focus of tinnitus research. The mechanism prevailing as responsible for triggering tinnitus, occurring for example after an acoustic shock, was based on a cascade of events following peripheral damage. In this view, cochlear hair cell impairment is proposed to lead to deafferentation of acoustic nerve fibers, which in turn drives plastic changes within auditory circuits and cortical structures. Similar changes have been described in other sensory modalities, such as in the somatosensory cortex after amputation [7,8] or in the visual cortex following visual field loss [9,10]. In line with these findings, tinnitus perception was therefore suggested to result from maladaptive plasticity similar to the maladaptive plasticity observed in amputees with phantom limb pain [11]. However, functional MRI studies in tinnitus participants with hearing loss revealed that hearing loss, rather than tinnitus, was responsible for the plasticity in the auditory cortex [12], and that tonotopic map reorganization in the auditory cortex was not a causal factor of tinnitus [13,14]. These findings, alongside evidence of tinnitus existing without hearing loss altogether or existing despite cochlear nerve ablation, reignited the debate on the cortical representation of tinnitus. Nowadays, the idea that changes in brain regions outside the auditory pathway, in particular the somatosensory pathway, the emotional and the attentional systems, could trigger the development and maintenance of tinnitus is more widely accepted in the community [15]. Additional non-invasive human studies support the involvement of somatosensory pathways in tinnitus [16] and of attentional and limbic networks with possible interactions with the auditory network [17].

In a particular population of subjects professionally exposed to impulse noises, Job and colleagues showed that impulse noises affect middle-ear function and may play a role in the early stages of auditory fatigue encompassing tinnitus [18]. Further investigating the potential role of the middle ear in noise-induced tinnitus, this author sought for its cortical representation and found it in the parietal operculum (OpP) [19]. Furthermore, as tinnitus may have a masking effect on sounds, Job and colleagues investigated the capacity of detecting a target sound among regular sounds using an oddball paradigm in functional MRI. In a group of participants with tinnitus compared to controls, they found an activation in the parietal operculum which was correlated with the chronicity and intensity scores of tinnitus perception [20]. No differential activity was found in auditory regions. This led this group to suspect the parietal operculum to be a core region of a tinnitus network. Further connectivity studies between non-bothersome tinnitus participants and controls evidenced the presence of a differential connectivity from the OpP and a frontal region posterior to the frontal eye field [21]. These authors also used a strategy that bypassed the question of comorbidities completely, by studying healthy subjects perceiving transitory phantom sounds resembling tinnitus. By using click trains at specific frequencies it was proven possible to reliably induce auditory after-effects mimicking the tinnitus perception (when using a 30 per second stimulation). This perception of sound in the absence of a corresponding external source thus makes for a proxy of tinnitus. The central representation of this latter stimulus has been explored with a high-resolution fMRI method and compared with a stimulus inducing no after-effect, revealing an increased activity in the somatosensory cortex in the face area and in the right OpP [22]. The induced phantom perception did not reveal any activation in the auditory cortex.

The near-complete absence of literature on OpP with regard to tinnitus is surprising. One explanation relates to its vicinity with the primary auditory cortex (A1) (Figure 1). Due to the spatial smoothing and the spatial normalization of MR images, the fMRI activations in OpP could be mistakenly reported as activations of the primary auditory cortex or of the posterior insular cortex (post-Ins). Indeed, located on the opposite banks of the lateral sulcus, OpP, post-Ins and A1 are close to one another, OpP being more anterior and superior and post-Ins more medial and anterior than A1, respectively. In the human cortex, regions that lie on opposite banks of a main sulcus generally present mirror organization, with mirror somatotopy around the central sulcus for primary motor on the anterior side and primary sensory on the posterior side, with mirror retinotopy around the calcarine sulcus with inferior part of a hemifield above the sulcus and the superior part below the sulcus. Up to now, no such mirror tonotopy has been observed in human areas opposite to A1. To our knowledge, a single study reported a tonotopic organization in the insula, but in a rodent model [23].

Considering these elements, we propose a review of the current knowledge about the parietal operculo–insular region to interrogate its involvement in tinnitus perception. We review findings at different scales: first, the microscopic level from multi-unit neuronal recordings in awake as well as retrograde tracing in monkeys up to post-mortem invasive explorations in humans; second, the mesoscopic level using intracerebral explorations in humans and third, at a macroscopic level as described in the MRI literature. The reviewed elements suggest the existence of a multimodal integration behavior in the target regions, namely somatosensory, auditory, more precisely in the antero-medial part of the OpP, coherent with the subregion of the Jülich atlas named OP3. Finally, we hypothesize that a dysfunction of this opercular region could be related to tinnitus perception as was observed in previous studies, with potential for new avenues for future treatments.

## 2. Microscopic Scale

Regions around the lateral sulcus were studied in non-human primates using single and multi-unit recordings. They include the auditory cortex and its subdivisions [24], the somatosensory cortex [25], the insula [26], the gustatory cortex [27]. More recently, authors demonstrated the presence of a multisensory integration area, just caudomedial to the primary auditory cortex with robust somatosensory and auditory co-representation [28] and in the most medial regions of SII of awake macaque monkeys, sensory and auditory and visual inputs [29] supporting the presence of a multisensory integration area, in the vicinity of the primary auditory and the primary sensory areas. Additional support for this view comes from injection of retrograde tracers to directly explore any potential links between unimodal areas. A study on marmosets provided evidence of specific projections linking areas of different modalities [30]. A multisensory cortical region adjacent to the posterior tip of the lateral sulcus projecting to auditory, somatosensory and visual motion areas was found. Other studies in macaque monkeys found auditory–somatosensory integration areas in the caudal belt of the auditory cortex [31]. Throughout evolution, the amount of cortex in the lateral sulcus has varied greatly across primates, from a rather shallow fissure in some prosimians to a deep fissure with a large fundus, the insula. The upper bank of the lateral sulcus includes the operculum and is largely devoted to somatosensory areas, while the lower or temporal bank is associated with auditory areas. In the greatly enlarged human operculum and insula, compared to those in apes and other primates, new regions have emerged, outgrowing the areas described in the non-human primate literature as involved in processing taste, pain, temperature, touch and internal states [32].

In humans, investigations of the opercular and insular regions began with the pioneering brain mapping studies performed by Brodmann at the beginning of the 20th century. Based on cytoarchitectony in the cortical layer and on homologies with other mammalian species, he was able to describe up to 52 different cortical areas. In the opercular part of the human cortex, he defined a single region, the area 43, located on the lateral side of the cortex. However, the masked part of the operculum remained undescribed. In the insula, he did not find the three respective homologue areas of the insular cortex of old-world monkeys. Instead, he described the anterior and the posterior parts of the insula, without providing any corresponding nomenclature, and the parainsular area 52 located between the posterior insula and the temporal area 41. In recent years, a remapping of the whole human brain was undertaken by the group led by K. Zilles in Jülich, providing a 3D probabilistic atlas of the human brain based on cytoarchitectony and receptor mappings [33]. This atlas contains the probabilistic maps derived from cytoarchitectonic studies of over 200 areas of the human brain including cortical areas and subcortical nuclei. The operculum presents nine subdivisions, five in the frontal operculum and four in the parietal operculum [34,35]. The lateral parts of the OpP contain OP1 posteriorly and OP4 anteriorly while the medial parts contain OP2 posteriorly and OP3 anteriorly. The insula is subdivided into 16 regions, 3 in the agranular part, 3 in the granular part and 10 in the dysgranular part. Their respective probabilistic locations are displayed in Figure 2 derived from the online interactive viewer of the Human Brain Project (https://interactive-viewer.apps.hbp.eu/, 1 December 2021). Additionally, two periauditory regions at the interface between the insula and Heschl’s gyrus, have been recently documented [36]. The functional evidence provided seems to advocate for the existence of junction regions presenting characteristics of both adjoining cortices and allowing a transition from early processes in the core auditory, towards insular integration.

These post-mortem studies provide very direct insight into brain structure at multiple scales. However, they build upon small datasets of ten post-mortem brains, which may be problematic in regions, such as Heschl’s gyrus, where individual variability is high.

## 3. Mesoscopic Scale

Invasive electrophysiological measurements provide a complementary brain investigation approach, with increased scalability potential. It provides the closest insight into in vivo human brain function combining high temporal and spatial resolution. The trade-off resides in the discrete spatial sampling dictated by the location of implantation chosen, based on clinical requirements: for obvious ethical considerations, no healthy subjects are included in invasive studies; therefore, functional brain mapping derived from intracerebral recordings is a fringe benefit from pathological brain exploration [37]. These issues are compensated by the large number of intracranial recordings, owing to their integration in the clinical investigation of prevalent conditions such as epilepsy. For example, the F-TRACT database has been recently constituted to centralize the direct electrical stimulations (DES) and stereoelectroencephalographic recordings of multiple areas of the human cortex, hippocampus and amygdala of several hundreds of subjects. The associated F-TRACT atlas provides functional connectivity information, which is oriented and direct according to the dynamical properties of the stimulation and recording data [38,39].

In this section, we first review the data from intracerebral electrical recordings of the insulo-opercular regions. Then, we consider the behavioral impacts of direct electrical stimulation paradigms and finally we review the state of the art provided by intra cortical connectivity of these regions.

### 3.1. Intracerebral Recordings of the Operculo–Insular Cortex

The involvement of the posterior insula in response to auditory stimuli has been addressed using intracerebral EEG (iEEG). Zhang and colleagues [40] collected iEEG data from epileptic patients, as well as functional MRI (fMRI) and Diffusion Tensor Imaging (DTI) in both epileptic and control subjects, listening to emotional and non-emotional stimuli, in order to determine the roles of the different subdivisions of the insula. Recording sites within the posterior insula responded to a wide range of acoustic stimuli, irrespective of emotional content, in line with previous findings [41,42,43,44,45]. The study also highlighted strong connections between the posterior insula and the auditory cortex with a stronger connectivity in the right as compared to the left hemisphere, and suggested a shared role of sound feature extractions, namely the central frequency and amplitude envelope of acoustic stimuli.

Yet, with the iEEG technique, it is not possible to know whether the parietal operculum responds also to acoustic stimuli since, to the best of our knowledge, the literature does not report iEEG data in this region during auditory stimulation.

### 3.2. Perceptions Mapping in the Operculo–Insular Cortex Induced by DES

DES provides direct insight into the behavioral responses, induced by electrical impulses, reported by the patients involved in intracortical stimulation protocols. This technique provides a direct link between perception and anatomical location, as long as the emergence of perceptions occurs. It has been used in vivo to help delineate the precise boundaries of the human primary auditory cortex on Heschl’s Gyrus [46].

Fewer studies have addressed the perceptions induced by DES in the operculo–insular cortex. To our knowledge, only one study specifically targeted the operculum [47], two others focused on the insula [45,48] and a third one targeted both regions [49]. An additional study explored the auditory responses to DES with contacts in forty-two different regions including these two regions [50]. In Maliia’s publication [47] reporting DES explorations of the operculum, the authors report the existence in the OpP, not only of somatosensory responses, including pain, but also of diverse effects with an asymmetry between sides. In the left OpP, there were mainly motor, visceral, language and vegetative effects, while the right OpP elicited mainly pain and auditory responses. In Mazzola’s publication [48] reporting DES explorations of the insula, the authors showed that the predominant modality was somatosensory (including pain) but auditory responses were also evoked, mainly located in the posterior long gyrus of the insula. Descriptions of the evoked sounds were ‘sizzling’, ‘buzzing’ or ‘drumming’, a vocabulary also used by tinnitus sufferers. As other modalities were also reported such as vestibular, cardiovascular or gustatory, there is converging evidence for multimodal representations with clear spatial overlap between sensory modalities. However, no auditory responses were found following OpP stimulation in Yu’s publication [49]. One reason might be that the location of the electrodes was limited to the inferior part of the OpP, with a risk of sampling bias as is likely in DES studies, and of missed effects. The recent exploration with DES of regions producing auditory hallucinations and illusions within the brain brings new evidence for a large territory eliciting auditory responses [50]. Indeed, DES in up to 42 cortical regions, according to the Jülich atlas, could elicit auditory responses, with 27 for hallucinations and 36 for illusions with large overlap between them. Not only were primary and secondary auditory areas but also the posterior and the mid insula as well as the OpP were involved in the hallucination perceptions. In this study, auditory hallucinations were described as simple hallucinations (elementary sounds such as clicking, whistling, ringing, buzzing) or complex hallucinations (elaborate auditory phenomena such as music or voices), where most descriptions of simple hallucinations were in line with tinnitus experience. Taken together, these studies support the view that auditory and somatosensory modalities are not integrated in a single area, as could be suggested by animal studies, but are rather somehow overlapping in the posterior insula and the parietal operculum, as evidenced by perceptions similar to tinnitus.

### 3.3. Operculo–Insular Connectivity

The term ‘cortico–cortico evoked potential’ (CCEP) describes the pattern of responses to a stimulation applied on a cortical target measured from all the other electrode contacts placed stereotactically inside the brain. The aim of this approach is to enable simultaneous mapping of functionally synchronized regions and their anatomical connections [51].

These responses to brief electrical pulses mimic the physiological propagation of signals along axonal populations and manifest as modulations of cortical activity in the connected regions. After stimulation artifact correction [52], the first CCEP component is considered to reflect direct connectivity with the stimulated region. The investigations using this methodology are able to demonstrate causal interactions between brain areas in vivo, known as effective connectivity [53]. To infer reliable connectivity information however, large amounts of data are required, which is what the multicenter F-TRACT initiative strives to do. Ultimately, the goal of the F-TRACT project is to provide a large-scale functional atlas for each region of the main brain parcellations, at different resolution scales (https://f-tract.eu, version 1 December 2021). Using our latest internal version of the F-TRACT atlas including a total of 942 implantations, we examined connectivity patterns from the posterior operculum (OP1, OP2, OP3 and OP4) and two insular regions (Ig1 and Ig2) presenting borders with OpP and auditory areas, as defined in the Jülich atlas of brain parcellations. Stimulation contacts belonging to the same parcellation region were pooled together and their responses, binarized by statistical thresholding with z-score threshold = 5, were used to yield a connectivity probability (for details see [38,54]). Thresholding was performed by excluding regions where less than five stimulations were performed, or where recordings originated from fewer than five different subjects. Thus, a square matrix of connectivity was generated, representing the probability that a response be detected in one region, when stimulating from another.

In Figure 3, all connections of the parietal operculum subregions are displayed as red dots, with the radius varying with connectivity strength. The lines represent the connections that survive the thresholding, and the colormap is set to represent connectivity values. The probability matrices are not symmetrical due to afferent and efferent connections: this is considered by modulating the coloring of the connection if the nodes at the extremities of an edge have different values in the connectivity matrix. This represents the causal relationships between regions. It appears that parietal opercular subregions (OP1-OP4) are strongly inter-connected. They are also connected by efferences with frontal opercular subregions and with granular and dysgranular subcomponents of the insula within the same hemisphere. In addition, left and right OP1, OP2 and OP3, but not OP4, present efferent connectivity with somatosensory cortex ipsilaterally. Interestingly, connections from the OP1, OP2 and OP3 subregions to the temporal auditory regions of the superior temporal gyrus (TE1, TE2.1, TE2.2) were only found in the right hemisphere and only OP2 in the left one. Connections to subregions of the Heschl gyrus were found from OP1, OP2 and OP4 in the right hemisphere and from OP2 in the left hemisphere. This asymmetry between right and left hemisphere could be related to the asymmetry of the auditory cortex with left hemisphere more sensitive to speech and right hemisphere to melodic contents [55]. Other efferences from all OpP subregions were found to inferior parietal lobule and from bilateral OP1 and OP3 to frontal area 44. No interhemispheric connectivity was reported here probably because the CCEP amplitude decays with distance and thus the chance of observing an existing connection decreases with the distance between regions. However, lowering the probability threshold permitted to find connectivity with homotopic regions of the opposite hemisphere. Notably no connections with the limbic system could be observed. To summarize the connections from OpP to both auditory and somatosensory cortices, only right OP1, bilateral OP2 and right OP3 were evidenced, providing potential somatosensory–auditory integration properties to those regions.

CCEP in the human operculum has been published based on the data acquired in 31 patients [47]. The data of this group and others are collected in the database of the F-TRACT project. Here, only the connectivity from a given region that survives strict thresholding (at least 10 stimulations from at least 5 patients) are further considered, which may explain why some results described by these authors were not reproduced here. The fact that all the red dots, including some with probability close to 1, are not connected also reflects the strict thresholding applied in the process.

In Figure 4, all connections from the granular subregions Ig1 and Ig2 of the insula, which present borders with both the parietal operculum and the temporal cortex, are displayed with the same methodology and threshold as for Figure 3. These regions show strong connections with one another and with Ig3, as well as with dysgranular parts of the insula. They also present connections with the subregions of the OpP and with the subregions of the Heschl gyrus and the superior temporal gyrus. Additional connection was found with the motor area 4p and frontal area 44 from right Ig1. All but left Ig2 were connecting to the inferior parietal lobule.

The richness of the connections between OpP and the insula, the auditory and the somatosensory areas confirms that OpP can be considered as a multimodal region, potentially involved in sensory and auditory integration. However, since electrodes were not covering all cortical regions, additional connectivity pathways could remain unobserved.

## 4. Macroscopic Scale

While intracerebral recordings provide a form of ground truth for neural brain activity, their main drawback relates to the sparsity of the electrode implantations. The non-invasive findings, mainly related to functional MRI, provide whole brain explorations that complement the invasive findings. In addition, while operculo–insular connectivity patterns can be established using the F-TRACT database, the functional aspects of these regions are not, which is an issue that neuroimaging can contribute to alleviate. In this section, we first review the current knowledge on the functions and connectivity of the opercular and insular cortices. Then, we review its involvement in the tinnitus literature.

### 4.1. Integration of Auditory and Somatosensory Stimuli in the Operculo–Insular Cortex

If the parietal operculum integrates auditory and somatosensory modalities, then this region should be involved in studies related to bimodal integration. Several human activities require such integration processes, such as speech production or music playing, for instance. Indeed, the left parietal operculum was found to be involved together with the auditory and somatosensory areas when looking at the language network [56], with a location in OP4 also covering OP3 and extending to the insula. In music playing, concomitant activation of sensorimotor and auditory systems is required. In fMRI studies related to music, functional connectivity analysis suggested the parietal operculum to be a connector hub linking auditory, somatosensory and motor areas [57]. This region was further found to be more strongly connected to other cortices in musicians than in non-musicians [58] with an asymmetry between both hemispheres. Interestingly, in an fMRI study on feelings evoked by music, the authors contrasted the joy- and the fear-music decoding conditions and found a large activation pattern including the parietal operculum bilaterally and extending into the posterior insula [59]. Those authors further proposed that secondary somatosensory cortex, which covers the parietal operculum and encroaches on the posterior insula, was of particular importance for the encoding of emotion percepts. An auditory frequency discrimination task was also found to involve the parietal operculum bilaterally [60]. A recent study manipulating sung speech stimuli by filtering them either temporally or spectrally allowed the authors to determine the reason for the hemispheric asymmetry observed between the auditory cortices. Using fMRI, the authors showed that the neural decoding of speech and melodies were represented by activity patterns in the left and right auditory regions, respectively [55]. These fMRI studies support the view of an integrative function of the OpP with an asymmetry related to the spectral and temporal modulations of auditory stimuli. However, only a few of these studies attempted to further identify which of the subregions of the OpP were activated.

One issue with the functional study of the opercular cortex comes from its vicinity with the insula. For instance, the meta-analysis provided for the insula in the neurosynth database (https://www.neurosynth.org/analyses/terms/insula/, 1 December 2021) shows that this region encompasses not only the insula but also the medial part of the operculum, thus potentially medial opercular subregions. This is mainly a side effect of the preprocessing of the fMRI data which usually includes a spatial smoothing of up to 8-mm. Another issue comes from the popular inflated representation of activation on the brain. In such figures, the limit between the adjacent operculum and insular cortex is unclear. It is thus difficult to disentangle activities related to each of these regions. However, despite these methodological limitations, several functions are known to be encoded in the insula. As expected from DES experiments, the presence of a specific network corresponding to pain, known as ‘pain matrix’, with a core posterior insular region has been described by Wager and co-authors [61]. It is noteworthy that not only the posterior insula, but also the parietal operculum, were involved in this network, among other areas. The insula subserves a variety of functions in humans ranging from somatosensory, emotion processing to high-level cognition [62]. A meta-analysis of nearly 1800 functional neuroimaging experiments by Kurth and colleagues suggested the existence of four functionally distinct regions in the human insula: first a sensorimotor region located in the mid-posterior insula; second, a central olfacto–gustatory region; third, a socio-emotional region in the anterior–ventral insula; and fourth, a cognitive anterior–dorsal region [63]. Coherent with the findings from DES, the insula is also involved in auditory processing such as sound detection. Finally, the insula seems to be involved in speech production, probably through higher-order articulatory processes. Other functions reported in the literature include olfactory, gustatory, viscero–autonomic, and limbic function for the anterior insula, and auditory–somesthetic–skeletomotor function for the posterior insula [62], but these studies bear the aforementioned uncertainty on their location.

Lesion studies provide additional support for the involvement of the parieto-insular cortex in tinnitus perception. A recent review about strokes located in the insula shows a large heterogeneity of clinical presentations with differential symptoms according to the side of the lesions [64]. Not only are sensory dysfunctions reported mainly in the posterior part of the insula with a balance between left and right hemisphere, but auditory disturbances, such as sounds evoking tinnitus, are also found, predominantly in the right hemisphere, alongside other symptoms such as dysarthria or vestibular conditions. Given that insular strokes mainly result from middle cerebral artery segments infarctions, deficits reported by the patients could relate directly from the lesion site, but also indirectly from surrounding tissues where hypoperfusion occurs, or from connections with the insula. The involvement of the insula as well as the neighboring operculum could thus cause these symptoms. This study provides additional grounds to support multimodal integrative processing in the parietal operculo–insular cortex.

### 4.2. Functional and Structural Connectivity of the Parietal Operculum and Insular Cortex

Connectivity analysis using seed-based methods can further specify the functional networks each of these subregions belong to. Such studies based on resting-state fMRI have shown that OP4 is more closely connected to areas responsible for basic sensorimotor processing and action control, while OP1 is more closely connected to the parietal networks for higher order somatosensory processing [65]. Meta-analyses have shown that the cytoarchitectonic area OP2 is a core region for a human vestibular network with a predominance in the right hemisphere [66,67]. This vestibular involvement of OP2 is further supported by patients with chronic bilateral vestibular failure [68]. Finally, OP3 seems to be involved in oral somatosensory stimulation, comprising the gustatory network [69], and laryngeal cortical network involved in swallowing [70,71]. Its involvement was also described in laughing and tickling [72], in the central representation of the tympanic membrane and middle-ear muscles in response to 1 Hz mechanical pressure variation [19] and in relation to temporomandibular joint disorder [73]. Taken together, the stimuli involving orofacial muscles thus seem to be represented in the cortical region OP3. In the insula, different patterns of functional connectivity were found for the anterior and posterior parts, the former connecting with frontal and limbic regions, while the latter connected to the sensorimotor, the auditory and the visual cortices [40].

Data from the Human Connectome Project provided additional information about the functional and structural connectivity of the parietal operculum and posterior insula. Based on the parcellation described by Glasser and numerous colleagues [74], Baker and colleagues described high mutual interconnectivity within the parietal opercular regions and granular part of the insula [75]. In short, functional connections were found with the sensory motor network, the auditory network, components of the visual network, with the cingulate areas, with subregions of the parietal opercula, and with the superior posterior insula. Most structural connections were local, connecting parietal opercular subdivisions, granular and anterior parts of the insula, and the auditory area, while some presented long-distance connections with the sensorimotor cortex [75]. While the HCP describes the cortico–cortical connections, other authors described the connectivity with the basal ganglia, the amygdala and the hippocampus showing a rich insular connectivity pattern with subcortical structures [76].

### 4.3. Encoding of Tinnitus in the Parietal Operculo–Insular Cortex

Tinnitus is a percept with limited temporal fluctuations and a predominant spectral component, ranging from pure tones to narrowband and broadband perceptions It has recently been established that spectral modulations of perceived sounds are processed preferentially in the right hemisphere, and temporal modulations in the left [55]. Based on these findings, tinnitus can be expected to be represented more predominantly in the right hemisphere than in the left.

If the tinnitus frequency band is partially masking other sound frequencies, then the discrimination of sound might be modified. Based on this hypothesis, an fMRI study was designed to explore the capability of subjects with tinnitus following acute acoustic trauma without comorbidities and control participants to discriminate auditory target stimuli as compared to standard stimuli, in an oddball paradigm [20]. The oddball task consisted of three auditory stimuli lasting 130 ms: standard stimuli (a sound with frequencies increasing linearly from 250 to 1000 Hz, occurring in 80% of cases, *n* = 348), target stimuli (a sound with frequencies decreasing linearly from 1000 Hz to 250 Hz occurring in 10% of cases, *n* = 48), and novel stimuli (different noises, such as onomatopoeia sounds found in cartoons, occurring in 10% of cases, *n* = 48). Importantly, although audiograms were different between control participants and tinnitus participants in frequencies above 2000 Hz, these differences were not significant in the frequency range of the auditory detection task, ruling out the influence of the audiometry in the results. Strong auditory attention is required to detect the difference between target stimuli and standard stimuli, a task that was found to be easier to perform for control participants and harder to perform for tinnitus participants. The authors found a set of regions differentially involved between the two groups, in particular in the parietal operculum of the right hemisphere. According to the Jülich atlas, its location is the border between OP1, OP2 and OP3 subregions. Moreover, in this region they showed that the differential activity in tinnitus subjects was increased with increasing tinnitus periodicity and handicap. As presented above in this review, this region is activated by movements of the orofacial muscles, and in particular the middle ear muscles [19,77]. One of the interests in this particular study comes from the inclusion criteria for the tinnitus group of subjects. They all presented a same etiology: chronic tinnitus following acoustic trauma, and none of them had bothersome comorbidities. In this case, the overactivity reported is probably related to tinnitus percept.

How to explain the involvement of this opercular region? The acoustic trauma likely generates a mechanical trauma of the middle-ear muscles, leading to abnormal excitability of middle-ear muscle spindles. Given that muscle spindles are related to proprioception, which presents a central representation in the parietal operculum [78], the hypothesis is that a hyperactivity is produced in the parietal opercular subregion corresponding to the middle-ear and interpreted by the cortex as an auditory percept. The fMRI study about temporomandibular joint disorder and its occlusion therapy [73] evidenced an overactivity in the very same location as in Job’s 2012 study. As tinnitus is known to be a comorbidity for temporomandibular joint disorders [79], the mediating role of the OP3 subregion of the parietal operculum might be considered. Since, as mentioned previously, OP3 seems to be an integration area for auditory and somatosensory representation of the orofacial muscles, and given that stimulation in this area, as observed by direct electrical stimulation, can elicit auditory hallucination, we could hypothesize that a dysfunction of these muscles following noise-induced fatigue may lead to tinnitus perception. This hypothesis may further explain why a majority of tinnitus sufferers are able to modulate their tinnitus perception by moving their face and neck [80,81].

Another original method is of interest in the study of tinnitus perception per se. It consists of inducing a transitory tinnitus sound in control subjects without tinnitus. This strategy allows the intrasubject comparison of conditions with tinnitus percept vs. no tinnitus percept, which presents the advantage of using a paired *t*-test at group level analysis, which is more sensitive to small differences than the two-sample *t*-test used in group comparisons. This was achieved with a train of click sounds at 30 Hz, which generated a tinnitus like after-effect, vs. a train of click sounds at 8 Hz which did not generate any after-effect. Specific vibration rates have been found to induce kinesthesic illusions in skelettal muscles at about 70 Hz. The tensor tympani and stapedius are capable of conveying proprioceptive informations at specific vibration rates around 30 Hz [22]. Thus, following 30 Hz click trains, an auditory–somatosensory integration could produce a tinnitus-like perception. An fMRI study was performed with high spatial accuracy and low scanner noise in the fMRI acquisition sequence to allow the perception of the auditory after-effect. It provided two small cortical foci of activation, one in the somatosensory cortex and the other one in the parietal operculum OP3 of the right hemisphere [22].

In the tinnitus literature, only a few studies reported the parietal operculum involvement. The large heterogeneity in the recruited participants for non-invasive studies with different comorbidities, different etiologies and auditory characteristics is likely responsible for findings going undetected under the radar. Another issue relates to the noisy environment of the MR scanner during fMRI acquisitions that limit the auditory studies. Another reason relates to the gross spatial resolution that would gain from improving accuracy and from including the recent atlases based on post-mortem cytoarchitectony Jülich or based on the in vivo data of the HCP. We advocate here for improvements in the methodology to better address the central representation of tinnitus and to better evaluate the therapeutic interventions that are currently under development.

### 4.4. Functional and Structural Connectivity with the Parietal Operculum in Tinnitus

To study the specific connectivity related to tinnitus perception while excluding confounding factors, two functional connectivity studies using resting-state fMRI, including only tinnitus subjects not impacted by comorbidities, were reported in the literature. The first one performed between non-bothersome tinnitus participants and the control group, investigating between-group differential connectivity between 58 seed regions and the whole brain, the seeds being chosen in the main functional networks in the default mode, attention, auditory, visual, somatosensory, and cognitive networks [5]. As the study could not elicit any significant differences between both groups, the authors suggested the non (or only poor) involvement of this pathways in the tinnitus perception. The second functional connectivity study with resting state fMRI explored more specifically the connectivity with the parietal operculum OP3 [21]. The authors found an increased connectivity between the right OP3 and two mirror regions of the dorsal prefrontal cortex, thought to correspond to the human homologue of the premotor ear-eye field bilaterally, and the inferior parietal lobule involved in proprioception.

Studies of structural connectivity associated with tinnitus present large discrepancies across the literature [82,83]. To the best of our knowledge, there exists only one study comparing participants with non-bothersome tinnitus to age-matched controls, using diffusion MRI and a crossing-fibers model that resolve more accurately the main 27 white matter tracts [84]. In this study, the authors highlighted white matter changes underneath the superior parietal cortex in tinnitus participants in a location supporting the implication of an auditory–somatosensory pathway in tinnitus perception. They also specifically investigated the acoustic radiations, that are challenging to model, but could not find any significant differences between groups.

## 5. Perspectives for Treatment of Tinnitus

The reviewed studies at different scales of investigation portray the parietal opercular region, and more precisely OP3, as an integrative area for auditory and somatosensory information from the orofacial muscles. Highly connected to the different subregions of the parietal operculum and of the posterior insula, it seems to be involved in the tinnitus perception per se induced by noise such as acoustic trauma or rifle impulse noise, in particular in the right hemisphere. It may thus be a good candidate for therapeutic interventions.

To date, many therapeutic strategies have been proposed to alleviate tinnitus, for which neuroimaging may provide a form of objective evaluation. They can be classified in four main methods. A first group involves specific acoustic stimulation paradigms. This is the case of dedicated hearing aids, sound therapy or music therapy. A second group relates to brain stimulation, either non-invasive such as repetitive Transcranial Magnetic Stimulation (rTMS) or transcranial Direct Current Stimulation (tDCS), or invasive such as deep brain stimulation (DBS). A third group relates to peripheral bimodal stimulation such as bimodal somatosensory and auditory stimulation, or autonomic and auditory stimulation. The fourth group pertains to the cognitive controls of tinnitus such as mindfullness-based therapy, tinnitus retraining therapy [85], or neurofeedback. A small, but growing, number of studies present longitudinal neuroimaging investigations to objectify the changes induced by the therapeutic intervention while correlating with tinnitus perception scores.

In the therapies based on auditory stimulation strategies, the impact of 6-month use of hearing aids in tinnitus subjects was evaluated by imaging metabolic glucose consumption with PET [86]. The authors found a pattern of increased and reduced glycolitic metabolism in many regions throughout the cortex. The neural correlates of sound therapy were evaluated by a single group, by measuring variation of grey matter thickness [87], of white matter volume [88] and amplitude of the low frequency fluctuations [89]. These studies found different modifications, however without showing a clear pattern of coherent results. Finally, the effects of the Heidelberg neuro-music therapy were investigated through measures of grey matter volume in acute tinnitus subjects [90]. These authors found significant modifications in different brain areas including the right parietal operculum following music therapy. Two different interpretations can arise from this music-based program: an OpP increased GM density related to tinnitus improvement or a musical training effect.

In the therapies based on brain stimulation, rTMS with a target above the left temporal lobe or above the left dorsolateral prefrontal cortex was proposed with only moderate efficacy with high interindividual variability in treatment response [91]. The present review indicates that the right parietal operculum could be considered as a new target for stimulation, but given its depth below the cortical surface, dedicated coils should be introduced in this case. The non-invasive tDCS has also been proposed as a therapeutic tool over the right dorsolateral prefrontal cortex, in particular in its high-density version that allows a finer accuracy of the target [92] with a significant improvement. Further investigations with neuroimaging tools are required, to better understand the neural correlates of this therapy. Invasive deep brain stimulation has recently been proposed with a target within the caudate nucleus which showed moderate efficacy, but other targets along the auditory pathways such as medial geniculate body or in the limbic networks are also proposed [93]. To date, invasive or non-invasive stimulations in the right operculum OP3 might be considered as a new avenue to provide relief from chronic tinnitus perception.

Therapies based on bimodal stimulation seem promising. The central effects of transcutaneous vagus nerve stimulation (tVNS) at different sites of the outer ear were explored by functional magnetic resonance imaging (fMRI) in patients with tinnitus. Targeting the tragus, a deactivation pattern of a large bilateral region encompassing both auditory areas as well as the parietal operculum was observed [94]. Combined with auditory stimulation, the tinnitus perception was alleviated in tinnitus subjects. It is postulated that the auditory–somatosensory integration process is disrupted during tVNS resulting in a suppression of tinnitus [95]. Recently Conlon and co-authors proposed a joint somatosensory–auditory stimulation, targeting the tongue with light electrical impulses and using sound stimulations to the ear, to modify the central integration process of these modalities [96]. The rationale for bimodal stimulation, is that it may counteract dysfunctional auditory–somatosensory interaction (leading to long term potentiation) in the dorsal cochlear nucleus. In the present review, we focused on the neocortex. However, the pathway to the neocortex involves lower-level structures. The proprioceptive information from the middle ear is conveyed through the trigeminal innervation and possibly by the non-lemniscal pathway of the brainstem. As a node of the non-lemniscal pathway, the dorsal cochlear nucleus receives inputs from the trigeminal nucleus [97], and we could hypothesize that it receives proprioceptive information. A crosstalk between auditory and somatosensory modalities has been shown to take place with the auditory pathway in the dorsal cochlear nucleus, as well as in the non-tonotopic medial geniculate body, where auditory–somatosensory integration processes take place [15,98]. Whether the lower-level multimodal integration is reflected in the multimodal integration OpP remains an open question. While encouraging clinical results were found, with strong interindividual variability, the central mechanisms induced by this therapy would benefit from neuroimaging explorations studies to objectify these results.

Finally, cognitive methods have been proposed. Habituation is a core mechanism in tinnitus retraining therapy and has been recently investigated using quantitative EEG [99]. The authors found an increased EEG power in the alpha 1 band in the right insula. Given the uncertainties provided by the source reconstruction methods used in this EEG study, the region encompasses also the right operculum and the right temporal pole. Mindfullness-based cognitive therapy has been proposed to reduce tinnitus severity. Zimmerman and colleagues could find functional connectivity modifications in different networks of the brain [100] but disentangling between those related to psychological modifications and those related to tinnitus perception remains an issue. Neurofeedback methods based on fMRI have recently been proposed to alleviate tinnitus by decreasing the neural activity in the auditory cortex [101]. The effect of neurofeedback of the auditory cortex revealed a pattern of regions where the activity is decreased, in particular in the right operculum [102], with no impact on the tinnitus perception, however.

## 6. Conclusions

In this review, we examined the evidence pertaining to an operculo–insular cross-talk hypothesis explaining tinnitus perception as an erroneous integrative process between multiple sensory inputs, mainly somatosensory and auditory. First and foremost, we review the current knowledge about subregions in the parietal operculum/posterior insula that present a dense tissue of connections and strong connectivity with auditory and somatosensory areas in normal brain function. These results provide the grounds to support the idea of a multiple sensory integration in this region. In the tinnitus perception literature, hyperactivity has been observed in the parietal operculum, suggesting dysfunctional integration properties. In the specific case of acoustic trauma tinnitus, the pathway leading to dysfunctional integration is proposed to be mediated by the middle ear as somatosensory input and inner ear as auditory input. Finally, the most encouraging therapeutic avenues today combine auditory and somatosensory stimulation, which may succeed in restoring functional multisensory integration, and lead to tinnitus alleviation.

## Figures and Tables

**Figure 1 brainsci-12-00334-f001:**
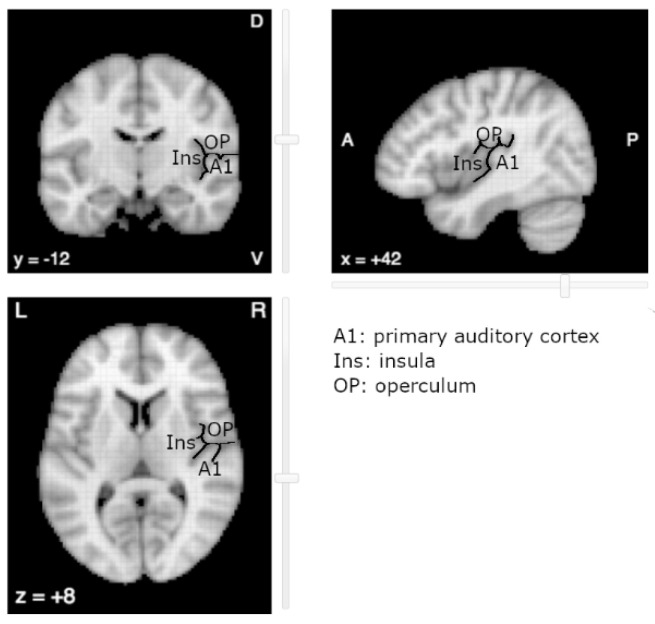
The auditory area (A1), parietal operculum (OP) and posterior Insula (Ins) lie on opposite banks of the lateral sulcus (underlined in black). Image derived from neurosynth.org (last accessed on 1 September 2021).

**Figure 2 brainsci-12-00334-f002:**
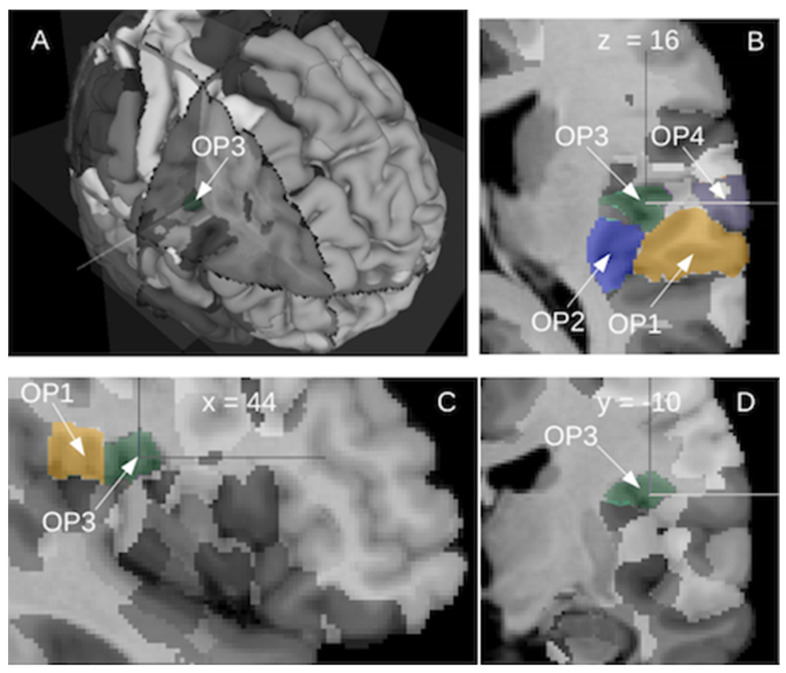
The posterior operculum of the right hemisphere shows four cytoarchitectonic subregions as defined by the Jülich group: lateral posterior OP1 (camel color), lateral anterior OP4 (grey), medial posterior OP2 (blue) and medial anterior OP3 (dark green). Different views: (**A**) 3D, (**B**) horizontal view, (**C**) sagittal view and (**D**) coronal view. Image derived from (https://interactive-viewer.apps.hbp.eu/, in 1 September 2021).

**Figure 3 brainsci-12-00334-f003:**
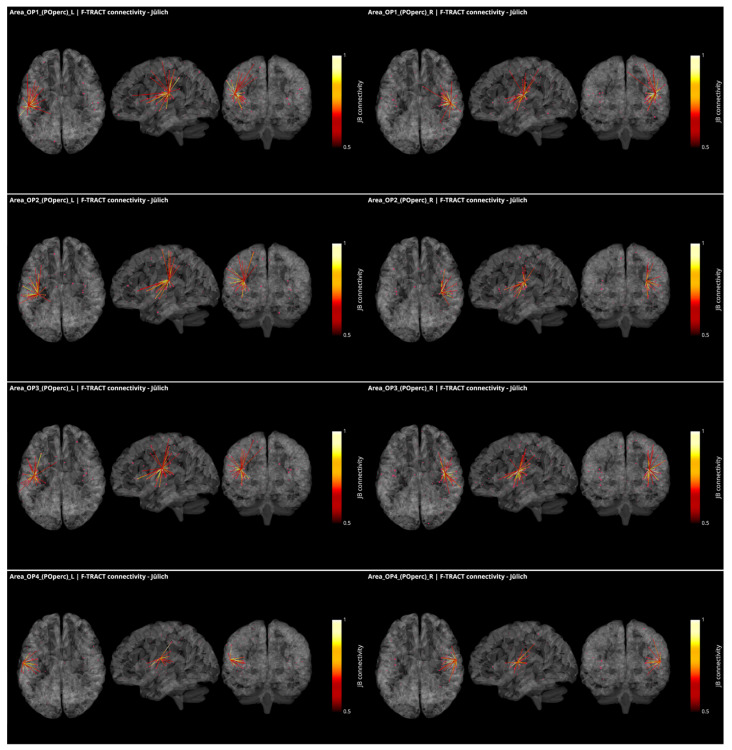
Intracortical connectivity of the four subregions of the parietal operculum from Jülich (‘JB connectivity’ stands for Jülich Brain connectivity). From top to bottom: area OP1, area OP2, area OP3 and area OP4. Left column for the subregions of the left hemisphere and right column for the subregions of the right hemisphere. Image derived from the data of the F-TRACT project (https://f-tract.eu), version of 1 December 2021.

**Figure 4 brainsci-12-00334-f004:**
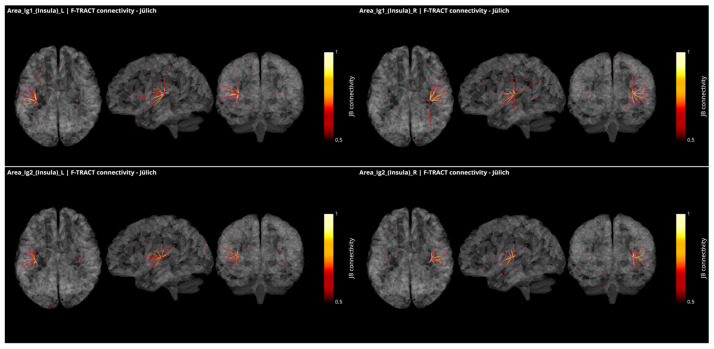
Intracortical connectivity of the two granular subregions of the insula. From top to bottom: area Ig1 and area Ig2. Left column for the subregions of the left hemisphere and right column for the subregions of the right hemisphere. Image derived from the data of the F-TRACT project (https://f-tract.eu), version of 1 December 2021.

## Data Availability

Data used for CCEP analysis can be found on the F-TRACT website (https://f-tract.eu/atlas/, 1 December 2021).

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
