# Peer review of "Tinnitus Perception in Light of a Parietal Operculo–Insular Involvement: A Review"

_brainsci, 2022, doi:10.3390/brainsci12030334_

Round 1

Reviewer 1 Report

This is an interesting manuscript, which presents a largely overlooked area of tinnitus neuroscience, namely the potential involvement of parietal operculum in tinnitus generation. The three key cited studies are from the authors’ own research group; two published studies implicate a specific region of parietal operculum in tinnitus due to acoustic trauma, and in transient tinnitus due to 30 Hz click trains. One submitted manuscript purportedly associates electrical stimulation of this area with simple and complex auditory hallucinations. The remainder of the study considers the structural and functional properties of parietal operculum, alongside the closely-related posterior insula, potential involvement in tinnitus, and methodological reasons for why activity in this region may have been erroneously attributed to nearby cortical areas in some previous literature.
I think this manuscript will be an interesting and valuable addition to the tinnitus literature, and I am broadly supportive of publication. I have certainly learned from reading it, and I think the key involvement of this area is plausible, and worthy of further ongoing research attention. However, I do have a number of points which addressing will improve the usefulness, robustness and accessibility of the manuscript.

Major points:
The case seems well-made that parietal operculum is an important area for audition, particularly I multimodal contexts. However, it is so far less clear that it is particularly distinct from the posterior insula in its functional roles. If the authors intend to convey such distinctions, they might include a section specifically contrasting posterior insula from parietal operculum. That said, they do go on to make a case for parietal operculum, specifically, in tinnitus, so the present point applies more to general auditory and multimodal roles.
Line 439: “If tinnitus frequency band is partially masking other sound frequencies then the discrimination of sound might be modified. Based on this hypothesis, a fMRI study was designed to explore the differential capability of tinnitus subjects without comorbidities”. This cited study [20] is a key piece of evidence for the potential role of parietal operculum in tinnitus. With this in mind, can the authors say a lot more here about the study? For instance, what type of auditory discrimination (frequency, intensity, etc.), what type of oddball in what type of paradigm? Also, the cited study is of tinnitus occurring in acute acoustic trauma. Finally, there seem to be group differences in audiometric thresholds between the three groups contrasted in that study, which needs to be considered when interpreting results.
Line 484: “different comorbidities, different etiologies and auditory characteristics”. I agree with this statement. However, the authors should also perhaps discuss that OP3 involvement might be specific to certain aetiologies of tinnitus (and specific mechanisms of transient tinnitus induction).
Line 520: “, it seems to be involved in the tinnitus perception per se”. I think this is something of an overstatement. The authors make a compelling case that it seems to be involved in tinnitus due to acute acoustic trauma, and transient tinnitus due to click trains. However, these may be just one route into tinnitus, rather than highlighting invariant common pathways, as the authors presently state/imply. On the same note, can the authors elaborate on the putative mechanism for how 30 Hz click trains induce transient tinnitus? This is really interesting, and highly relevant here in considering how to interpret findings of the study using this method.
Line 543: “Finally, the effects of Heidelberg neuro-music therapy were investigated through measures of the grey matter volume in acute tinnitus subjects [88]. These authors found significant modifications in different brain areas including the right parietal operculum following the music therapy” Surely this is hardly surprising, as parietal operculum is involved in musical listening, so this result could be considered a musical training effect?
Line 598: “the most encouraging therapeutical avenues today combine auditory and somatosensory stimulation” Very true. However, nowhere in this manuscript do the authors mention the standard rationale given for bimodal stimulation, which is that there is very well-characterised dysfunctional auditory-somatosensory interaction (leading to LTP) in the dorsal cochlear nucleus. The authors really should at least acknowledge this; whilst it does not render their claims of the importance of parietal operculum irrelevant by any means, it does mean that there are alternative or complementary explanations.
Related to the above, can the authors please discuss the impact that lower-level auditory-somatosensory interactions might have on activity in the parietal operculum multimodal area(s)? For instance, can any of what they have cited be explained as integrative processes carried forward from lower levels, or is there evidence to suggest this really is specifically a primarily cortical effect?
Minor points:
Line 365: “feelings evoked with music activated the parietal operculum” – The data cited presumably only refer to a correlation, so it would be best to phrase this claim as a correlation, rather than causation
Line 357: “and was found to be of particular importance for the encoding of emotion percepts”. This sounds like the interpretation of the authors of the cited paper. It would be better to describe what the results actually showed, instead of (or as well as) the interpretation of those results.
Line 372: “decoding of speech and melodies depends on activity patterns”. Again, this should be phrased as correlation rather than causation
Line 397: “Coherent with the findings from DES, the insula is also involved in auditory processing such as sound detection. Finally the insula seems to be involved in speech production, probably through higher-order articulatory processes. Other functions reported in the literature include olfactory, gustatory, viscero-autonomic, and limbic function for the anterior insula, and auditory-somesthetic-skeletomotor function for the posterior insula, but these studies bear the aforementioned uncertainty on their location”. Are there one or more citations to back up these claims?
Line 404: “Connectivity analysis using seed-based methods can further precise the functional…”. This is one example of sub-optimal grammar. There are too many such instances for me to list these individually in this review. The manuscript should be proof-read and corrected by somebody fluent in scientific English writing.
Line 411: The word “confirmed” should be “supported” or equivalent.
Line 435: “Tinnitus is a percept whose temporal fluctuations are almost absent while the frequency component is broadband, thus closer to melody than to speech. This is why tinnitus can be expected to be represented more predominantly in the right hemisphere than in the left.” I have a number of issues with this text. Firstly, the frequency component of tinnitus tends to range between tonal (i.e. as narrowband as is possible, perceptually) and narrowband noise, with broadband noise being less common. Secondly, melody is characterised by changing pitch, so it is unclear how melody is relevant here. There is presumably literature on the hemispheric representation of tinnitus, and asymmetries thereof, which should be cited if such claims are being made.
Line 459: “The fMRI study about temporomandibular joint disorder”. Please provide a reference here, as it is presently not clear which study the authors mean.
Line 502: “The second functional connectivity study with resting state fMRI explored more specifically the connectivity with the parietal operculum OP3 [21].” Reference 21 is a submitted manuscript on induction of hallucinatory percepts due to electrical stimulation, not a resting-state fMRI paper on tinnitus as cited.
Figure 1: It would be good to colour the exact position and extent of the parietal operculum to show its limits/borders
Figure 2: I think there are too many colours here, which makes it hard to interpret and orient oneself. Perhaps just colouring the parietal operculum areas would help, as the colouring of other areas obscures familiar brain regions which the reader will use to orient themselves.

Author Response

Responses to reviewer 1
This is an interesting manuscript, which presents a largely overlooked area of tinnitus neuroscience, namely the potential involvement of parietal operculum in tinnitus generation. The three key cited studies are from the authors’ own research group; two published studies implicate a specific region of parietal operculum in tinnitus due to acoustic trauma, and in transient tinnitus due to 30 Hz click trains. One submitted manuscript purportedly associates electrical stimulation of this area with simple and complex auditory hallucinations. The remainder of the study considers the structural and functional properties of parietal operculum, alongside the closely-related posterior insula, potential involvement in tinnitus, and methodological reasons for why activity in this region may have been erroneously attributed to nearby cortical areas in some previous literature.
I think this manuscript will be an interesting and valuable addition to the tinnitus literature, and I am broadly supportive of publication. I have certainly learned from reading it, and I think the key involvement of this area is plausible, and worthy of further ongoing research attention. However, I do have a number of points which addressing will improve the usefulness, robustness and accessibility of the manuscript.
We thank the reviewer for their consideration of our manuscript for publication.
We add here that english editing has been performed throughout the manuscript following one of the reviewer’s recommendation.
Major points:
The case seems well-made that parietal operculum is an important area for audition, particularly I multimodal contexts. However, it is so far less clear that it is particularly distinct from the posterior insula in its functional roles. If the authors intend to convey such distinctions, they might include a section specifically contrasting posterior insula from parietal operculum. That said, they do go on to make a case for parietal operculum, specifically, in tinnitus, so the present point applies more to general auditory and multimodal roles.
We thank the reviewer for this comment. Indeed, the remark of the reviewer is interesting. However, the objective of this study being to draw attention to the parietal operculum and to its connections, in application to tinnitus, we chose to kept focus on the tinnitus.
Line 439: “If tinnitus frequency band is partially masking other sound frequencies then the discrimination of sound might be modified. Based on this hypothesis, a fMRI study was designed to explore the differential capability of tinnitus subjects without comorbidities”. This cited study [20] is a key piece of evidence for the potential role of parietal operculum in tinnitus. With this in mind, can the authors say a lot more here about the study? For instance, what type of auditory discrimination (frequency, intensity, etc.), what type of oddball in what type of paradigm? Also, the cited study is of tinnitus occurring in acute acoustic trauma. Finally, there seem to be group differences in audiometric thresholds between the three groups contrasted in that study, which needs to be considered when interpreting results.
Indeed, the description of this work requires further details. We modified the following to the text (l.485 in the revised version):
«Based on this hypothesis, an fMRI study was designed to explore the capability of subjects with tinnitus following acute acoustic trauma without comorbidities and control participants to discriminate auditory target stimuli as compared to standard stimuli having the same frequency content, in an oddball paradigm [20]. The oddball task consisted in three auditory stimuli lasting 130 ms : standard stimuli (a sound with frequencies increasing linearly from 250 and 1000 Hz, occurring in 80% of cases, n=348), target stimuli (a sound with frequencies decreasing linearly from 1000 Hz to 250 Hz occurring in 10% of cases, n=48), and novel stimuli (different noises, such as onomatopoeia sounds found in cartoons, occurring in 10% of cases, n=48). Importantly, although audiograms were different between control participants and tinnitus participants above 2kHz, these differences were not significant in the frequency range of the auditory detection task, ruling out the influence of the audiometry in the results. A strong auditory attention is required to detect the difference between target stimuli and standard stimuli, a task that was found easier to perform for control participants and harder to perform for tinnitus participants. »
Line 484: “different comorbidities, different etiologies and auditory characteristics”. I agree with this statement. However, the authors should also perhaps discuss that OP3 involvement might be specific to certain aetiologies of tinnitus (and specific mechanisms of transient tinnitus induction).
We thank the reviewer for this comment. Indeed, the evidence pertaining to OP3 in this review essentially relates to noise induced tinnitus that could lead to middle ear muscles dysfunction. We added this precision in the text (l.524 in the revised version).
« we could hypothesize that a dysfunction of these muscles following noise-induced fatigue may lead to tinnitus perception »
Line 520: “, it seems to be involved in the tinnitus perception per se”. I think this is something of an overstatement. The authors make a compelling case that it seems to be involved in tinnitus due to acute acoustic trauma, and transient tinnitus due to click trains. However, these may be just one route into tinnitus, rather than highlighting invariant common pathways, as the authors presently state/imply. On the same note, can the authors elaborate on the putative mechanism for how 30 Hz click trains induce transient tinnitus? This is really interesting, and highly relevant here in considering how to interpret findings of the study using this method.
Good point. This is the case mostly in noise-induced tinnitus, where the middle-ear compliance may be affected as shown in soldiers following rifle impulse noise [18]. We modified (l. 584 in the revised version):
« Highly connected to the different subregions of the parietal operculum and of the posterior insula, it seems to be involved in the tinnitus perception per se induced by noise such as acoustic trauma or rifle impulse noise, in particular in the right hemisphere.»
and we also provide a putative mechanism for how 30 Hz click trains induce transient tinnitus, an explanation that was previously published in [22] (l.516 in the revised version):
« Specific vibration rates have been found to induce kinesthesic illusions in skeletal muscles at about 70 Hz. The tensor tympani and stapedius are capable of conveying proprioceptive information at specific vibration rates around 30Hz [22]. Thus, following 30 Hz click trains, an auditory-somatosensory integration could produce a tinnitus-like perception. »
Line 543: “Finally, the effects of Heidelberg neuro-music therapy were investigated through measures of the grey matter volume in acute tinnitus subjects [88]. These authors found significant modifications in different brain areas including the right parietal operculum following the music therapy” Surely this is hardly surprising, as parietal operculum is involved in musical listening, so this result could be considered a musical training effect?
The reviewer is right. Two different interpretations can arise from this music program : an OpP increased GM density related to tinnitus improvement or a musical training effect. We added both interpretations in the text (l.613 in the revised version):
« Two different interpretations can arise from this music-based program : an OpP increased GM density related to tinnitus improvement or a musical training effect. »
Line 598: “the most encouraging therapeutical avenues today combine auditory and somatosensory stimulation” Very true. However, nowhere in this manuscript do the authors mention the standard rationale given for bimodal stimulation, which is that there is very well-characterised dysfunctional auditory-somatosensory interaction (leading to LTP) in the dorsal cochlear nucleus. The authors really should at least acknowledge this; whilst it does not render their claims of the importance of parietal operculum irrelevant by any means, it does mean that there are alternative or complementary explanations.
Thank you for this valuable suggestion. Following ref [94] of Coulon and coauthors, we added (l.643 in the revised version) :
« The rationale for bimodal stimulation, is that it may counteract a dysfunctional auditory-somatosensory interaction (leading to long term potentiation) in the dorsal cochlear nucleus. »
Related to the above, can the authors please discuss the impact that lower-level auditory-somatosensory interactions might have on activity in the parietal operculum multimodal area(s)? For instance, can any of what they have cited be explained as integrative processes carried forward from lower levels, or is there evidence to suggest this really is specifically a primarily cortical effect?
This is indeed a possible discussion. The role of lower-levels is not developed in this publication and we understand that it is useful to mention it. We thus added the following sentences (l.645):
« In the present review, we focused on the neocortex. However, the pathway to the neocortex involves lower-level structures. The proprioceptive information from the middle ear is conveyed through the trigeminal innervation and possibly by the non-lemniscal pathway of the brainstem. As a node of the non-lemniscal pathway, the dorsal cochlear nucleus receives inputs from the trigeminal nucleus [97], and we could hypothesise that it receives proprioceptive information. A crosstalk between auditory and somatosensory modalities has been shown to take place with the auditory pathway in the dorsal cochlear nucleus, as well as in the non-tonotopic medial geniculate body, where auditory-somatosensory integration processes take place [15]. Whether the lower-level multimodal integration is reflected in the multimodal integration OpP remains an open question.
Minor points:
Line 365: “feelings evoked with music activated the parietal operculum” – The data cited presumably only refer to a correlation, so it would be best to phrase this claim as a correlation, rather than causation
In fact, causation is not presented in the study. In the publication of Koelsch and co-authors, the contrast between joy- and fear-music elicited a large pattern of activation [59], that included the parietal operculum and the posterior insula. We better reported those resuts (l.384):
«  … in an fMRI study on feelings evoked by music, the authors contrasted the joy- and the fear-music decoding conditions and found a large activation pattern including the parietal operculum ... »
Line 357: “and was found to be of particular importance for the encoding of emotion percepts”. This sounds like the interpretation of the authors of the cited paper. It would be better to describe what the results actually showed, instead of (or as well as) the interpretation of those results.
Further in the same publication, Koelsch and co-authors suggested that « secondary somatosensory cortex, which covers the parietal operculum and encroaches on the posterior insula, is of particular importance for the encoding of emotion percepts ». This is not our interpretation of Koelsch’s paper but their proposition that we report here. To avoid possible misunderstanding, we modified the sentence including the previous modification in the following way (l.384 in the revised version):
«  Interestingly, in an fMRI study on feelings evoked by music, the authors contrasted the joy- and the fear-music decoding conditions and found a large activation pattern including the parietal operculum bilaterally and extending into the posterior insula [59]. Those authors further proposed that secondary somatosensory cortex, which covers the parietal operculum and encroaches on the posterior insula, was of particular importance for the encoding of emotion percepts. »  
Line 372: “decoding of speech and melodies depends on activity patterns”. Again, this should be phrased as correlation rather than causation
We modified in the following way (l.395):
« decoding of speech and melodies were represented by activity patterns ... »
Line 397: “Coherent with the findings from DES, the insula is also involved in auditory processing such as sound detection. Finally the insula seems to be involved in speech production, probably through higher-order articulatory processes. Other functions reported in the literature include olfactory, gustatory, viscero-autonomic, and limbic function for the anterior insula, and auditory-somesthetic-skeletomotor function for the posterior insula, but these studies bear the aforementioned uncertainty on their location”. Are there one or more citations to back up these claims?
We added the reference [62].
Line 404: “Connectivity analysis using seed-based methods can further precise the functional…”. This is one example of sub-optimal grammar. There are too many such instances for me to list these individually in this review. The manuscript should be proof-read and corrected by somebody fluent in scientific English writing.
The manuscript was proof-read accordingly.
Line 411: The word “confirmed” should be “supported” or equivalent.
Done
Line 435: “Tinnitus is a percept whose temporal fluctuations are almost absent while the frequency component is broadband, thus closer to melody than to speech. This is why tinnitus can be expected to be represented more predominantly in the right hemisphere than in the left.” I have a number of issues with this text. Firstly, the frequency component of tinnitus tends to range between tonal (i.e. as narrowband as is possible, perceptually) and narrowband noise, with broadband noise being less common. Secondly, melody is characterised by changing pitch, so it is unclear how melody is relevant here. There is presumably literature on the hemispheric representation of tinnitus, and asymmetries thereof, which should be cited if such claims are being made.
We agree that an ambiguity arises here. What is established, is that spectral modulation is preferentially processed in the right hemisphere (which explains why musical processing is preferentially performed in the right hemisphere – and not primarily because of the pitch modulation), and that temporal modulation is preferentially performed in the left. Based on the absence of temporal modulation of subjective tinnitus perceptions, we suggest that tinnitus may preferentially manifest in the right hemisphere, although this does not seem to have been specifically explored in the literature. Existing studies on tinnitus laterality refer to the laterality of the perception, not its central representation.
We propose the following modifications (l.477):
« Tinnitus is a percept with limited temporal fluctuations, and a predominant spectral component, ranging from pure tones to narrowband and broadband perceptions. It has recently been established that spectral modulations of perceived sounds are preferentially processed in the right hemisphere, and temporal modulations in the left [Albouy]. Based on these findings, tinnitus can be expected to be represented more predominantly in the right hemisphere than in the left. »
Line 459: “The fMRI study about temporomandibular joint disorder”. Please provide a reference here, as it is presently not clear which study the authors mean.
We added the reference [73]
Line 502: “The second functional connectivity study with resting state fMRI explored more specifically the connectivity with the parietal operculum OP3 [21].” Reference 21 is a submitted manuscript on induction of hallucinatory percepts due to electrical stimulation, not a resting-state fMRI paper on tinnitus as cited.
Sorry for the mistake. We added ref [83]
Figure 1: It would be good to colour the exact position and extent of the parietal operculum to show its limits/borders
Figure 1 is intended to display the respective locations of the auditory cortex, the insular cortex and the operculum with respect to the lateral sulcus. For a the exact positions of the subregions of the parietal operculum, figure 2 is better.
Figure 2: I think there are too many colours here, which makes it hard to interpret and orient oneself. Perhaps just colouring the parietal operculum areas would help, as the colouring of other areas obscures familiar brain regions which the reader will use to orient themselves.
Thanks for the remark. We put in grey-levels the colors of the regions not pertaining to the OpP, so that the 4 subregions OP1, OP2, OP3 and OP4 appear more clearly. New figure 2 is displayed l.188

Reviewer 2 Report
This review supports the idea that parieto-operculum-insular connections might represent a physiological basis for tinnitus and summarizes the current evidence on anatomy, physiology and fMRI studies on tinnitus. Also, it provides a very clear and complete model describing the principal hubs of tinnitus’s network. A sufficient emphasis is reported on the posterior insular cortex, in accord with lesion studies and fMRI studies. The review is well-conducted and the reference list is quite accurate. Style and grammar are adequate. I think that this review might be precious and useful, but I have some suggestions:
- The review provide evidence on fMRI data, brain connectome but lesion studies are no cited nor discussed. Of interest, lesion studies support the role of the posterior insula in tinnitus perception. Indeed, auditory disturbances have been described in patients with stroke confined to the insula with increased sensitivity to sounds or auditory allucinations sometimes similar to video games’ noise. I suggest to read and cite a very recent review on insular damage (Clinical presentation of strokes confined to the insula…2021), reporting precious data on the site and side of lesion. For example, it is reported that right damage is more often associated with auditory disturbances if compared to the left and also lesions in the anterior insula are reported in patients with auditory symptoms. Do the authors think that tinnitus was caused by the insular involvement per se or was due the operculum damage? These recent data should be widely discussed.
-I also suggest to underline and discuss if there is any evidence on lateralization of tinnitus from fMRI studies. If yes, what the authors hypothesize to explain this asymmetry?

Author Response

Response to Reviewer 2
This review supports the idea that parieto-operculum-insular connections might represent a physiological basis for tinnitus and summarizes the current evidence on anatomy, physiology and fMRI studies on tinnitus. Also, it provides a very clear and complete model describing the principal hubs of tinnitus’s network. A sufficient emphasis is reported on the posterior insular cortex, in accord with lesion studies and fMRI studies. The review is well-conducted and the reference list is quite accurate. Style and grammar are adequate. I think that this review might be precious and useful, but I have some suggestions:
Thank you for this positive feedback.
According to the recommendation of reviewer 1, we performed english editing throughout the manuscript that appear thus in red in the revised version
- The review provide evidence on fMRI data, brain connectome but lesion studies are no cited nor discussed. Of interest, lesion studies support the role of the posterior insula in tinnitus perception. Indeed, auditory disturbances have been described in patients with stroke confined to the insula with increased sensitivity to sounds or auditory allucinations sometimes similar to video games’ noise. I suggest to read and cite a very recent review on insular damage (Clinical presentation of strokes confined to the insula…2021), reporting precious data on the site and side of lesion. For example, it is reported that right damage is more often associated with auditory disturbances if compared to the left and also lesions in the anterior insula are reported in patients with auditory symptoms. Do the authors think that tinnitus was caused by the insular involvement per se or was due the operculum damage? These recent data should be widely discussed.
We thank the reviewer for their suggestions and for this pertinent citation. We added a citation and the following comment in the manuscript (l.426):
« Lesion studies provide additional ground for the involvement of the parieto-insular cortex in tinnitus perception. A recent review about strokes located in the insula shows a large heterogeneity of clinical presentations with differential symptoms according to the side of the lesions [64]. Not only sensory dysfunctions are reported mainly in the posterior part of the insula with a balance between left and right hemisphere but also auditory disturbances, such as sounds evocating tinnitus, that are predominantly in the right hemisphere and others symptoms such as dysarthria or vestibular. Given that insular strokes mainly result from middle cerebral artery segments infarctions, deficits reported by the patients could relate directly from the lesion site but also undirectly from surrounding tissues where hypoperfusion occurs or undirectly from connections with the insula. The involvement of the insula but also the contiguous operculum could thus cause these symptoms. This study gives additional ground for of multimodal integrative processing in the parietal operculo-insular cortex. »
-I also suggest to underline and discuss if there is any evidence on lateralization of tinnitus from fMRI studies. If yes, what the authors hypothesize to explain this asymmetry?
This is an interesting question. Yet, the relation between tinnitus lateralisation and hemispheric dominance remains an object of debate. We propose a hypothesis, independent of the side of tinnitus percept, based on evidence of preferential processing of spectral modulations in the right hemisphere, and temporal modulations in the left to explain its representation in the right hemisphere (l.477 in the revised version).
« Tinnitus is a percept with limited temporal fluctuations and a predominant spectral component, ranging from pure tones to narrowband and broadband perceptions. It has recently been established that spectral modulations of perceived sounds are processed preferentially in the right hemisphere, and temporal modulations in the left [55]. Based on these findings, tinnitus can be expected to be represented more predominantly in the right hemisphere than in the left. »
